# Reasoning About Physical Interactions with Object-Oriented Prediction and Planning

**Michael Janner**[†]**, Sergey Levine**[†]**, William T. Freeman**[‡]**, Joshua B. Tenenbaum**[‡]**,
Chelsea Finn**[†]**, & Jiajun Wu**[‡]

[†]University of California, Berkeley
[‡]Massachusetts Institute of Technology
`{janner,svlevine,cbfinn}@berkeley.edu`
`{billf,jbt,jiajunwu}@mit.edu`

## Abstract

Object-based factorizations provide a useful level of abstraction for interacting with the world. Building explicit object representations, however, often requires supervisory signals that are difficult to obtain in practice. We present a paradigm for learning object-centric representations for physical scene understanding without direct supervision of object properties. Our model, Object-Oriented Prediction and Planning (O2P2), jointly learns a perception function to map from image observations to object representations, a pairwise physics interaction function to predict the time evolution of a collection of objects, and a rendering function to map objects back to pixels. For evaluation, we consider not only the accuracy of the physical predictions of the model, but also its utility for downstream tasks that require an actionable representation of intuitive physics. After training our model on an image prediction task, we can use its learned representations to build block towers more complicated than those observed during training.

## 1 Introduction

Consider the castle made out of toy blocks in Figure 1a. Can you imagine how each block was placed, one-by-one, to build this structure? Humans possess a natural physical intuition that aids in the performance of everyday tasks. This physical intuition can be acquired, and refined, through experience. Despite being a core focus of the earliest days of artificial intelligence and computer vision research (Roberts, 1963; Winston, 1970), a similar level of physical scene understanding remains elusive for machines.

Cognitive scientists argue that humans' ability to interpret the physical world derives from a richly structured apparatus. In particular, the perceptual grouping of the world into objects and their relations constitutes *core knowledge* in cognition (Spelke & Kinzler, 2007). While it is appealing to apply such an insight to contemporary machine learning methods, it is not straightforward to do so. A fundamental challenge is the design of an interface between the raw, often high-dimensional observation space and a structured, object-factorized representation. Existing works that have investigated the benefit of using objects have either assumed that an interface to an idealized object space already exists or that supervision is available to learn a mapping between raw inputs and relevant object properties (for instance, category, position, and orientation).

Assuming access to training labels for all object properties is prohibitive for at least two reasons. The most apparent concern is that curating supervision for all object properties of interest is difficult to scale for even a modest number of properties. More subtly, a representation based on semantic

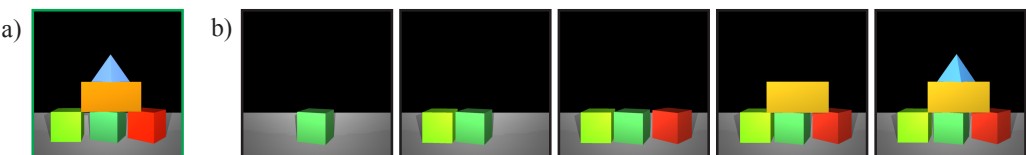

Figure 1: **(a)** A toy block castle. **(b)** Our method's build of the observed castle, using its learned object representations as a guide during planning.

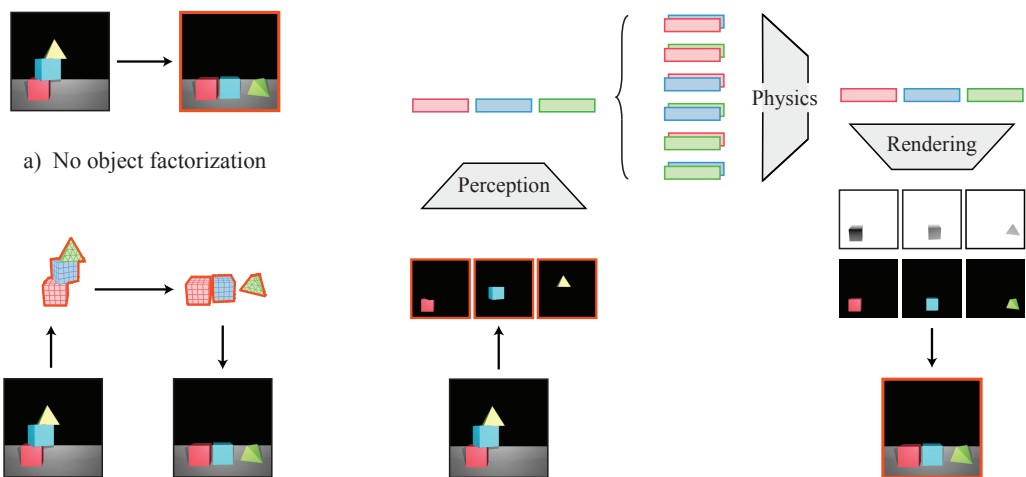

a) No object factorization

b) Object property supervision      c) O2P2: Object factorization without object property supervision

Figure 2: We divide physical understanding tasks into three distinct paradigms. **(a)** The first approach makes the fewest assumptions, posing prediction tasks as an instance of image-to-image translation. **(b)** The second uses ground-truth labels of object properties to supervise a learning algorithm that can map to the space of a traditional or learned physics engine. **(c)** O2P2, like (b), employs an object factorization and the functional structure of a physics engine, but like (a), does not assume access to supervision of object properties. Without object-level supervision, we must jointly learn a perception function to map from images to objects, a physics engine to simulate a collection of objects, and a rendering engine to map a set of objects back to a single composite image prediction. In all three approaches, we highlight the key supervision in orange.

attributes can be limiting or even ill-defined. For example, while the size of an object in absolute terms is unambiguous, its orientation must be defined with respect to a canonical, class-specific orientation. Object categorization poses another problem, as treating object identity as a classification problem inherently limits a system to a predefined vocabulary.

In this paper, we propose Object-Oriented Prediction and Planning (O2P2), in which we train an object representation suitable for physical interactions without supervision of object attributes. Instead of direct supervision, we demonstrate that segments or proposal regions in video frames, without correspondence between frames, are sufficient supervision to allow a model to reason effectively about intuitive physics. We jointly train a perception module, an object-factorized physics engine, and a neural renderer on a physics prediction task with pixel generation objective. We evaluate our learned model not only on the quality of its predictions, but also on its ability to use the learned representations for tasks that demand a sophisticated physical understanding.

## 2   OBJECT-ORIENTED PREDICTION AND PLANNING (O2P2)

In this section, we describe a method for learning object-based representations suitable for planning in physical reasoning tasks. As opposed to much prior work on object-factorized scene representations (Section 4), we do not supervise the content of the object representations directly by way of labeled attributes (such as position, velocity, or orientation). Instead, we assume access only to segments or region proposals for individual video frames. Since we do not have labels for the object representations, we must have a means for converting back and forth between images and object representations for training. O2P2 consists of three components, which are trained jointly:

- A **perception** module that maps from an image to an object encoding. The perception module is applied to each object segment independently.

- A **physics** module to predict the time evolution of a set of objects. We formulate the engine as a sum of binary object interactions plus a unary transition function.

- A **rendering** engine that produces an image prediction from a variable number of objects. We first predict an image and single-channel heatmap for each object. We then combine all of the object images according to the weights in their heatmaps at every pixel location to produce a single composite image.

A high-level overview of the model is shown in Figure 2c. Below, we give details for the design of each component and their subsequent use in a model-based planning setting.

## 2.1 PERCEPTION MODULE

The perception module is a four-layer convolutional encoder that maps an image observation to object representation vectors $\mathbf{O} = \{o_k\}_{k=1...N}$. We assume access to a segmentation of the input image $\mathbf{S} = \{s_k\}_{k=1...N}$ and apply the encoder individually to each segment. The perception module is not supervised directly to predict semantically meaningful properties such as position or orientation; instead, its outputs are used by the physics and rendering modules to make image predictions. In this way, the perception module must be trained jointly with the other modules.

## 2.2 PHYSICS MODULE

The physics module predicts the effects of simulating a collection of object representations $\mathbf{O}$ forward in time. As in Chang et al. (2016); Watters et al. (2017), we consider the interactions of all pairs of object vectors. The physics engine contains two learned subcomponents: a unary transition function $f_{\text{trans}}$ applied to each object representation independently, and a binary interaction function $f_{\text{interact}}$ applied to all pairs of object representations. Letting $\bar{\mathbf{O}} = \{\bar{o}_k\}_{k=1...N}$ denote the output of the physics predictor, the $k^{\text{th}}$ object is given by $\bar{o}_k = f_{\text{trans}}(o_k) + \sum_{j \neq k} f_{\text{interact}}(o_k, o_j) + o_k$, where both $f_{\text{trans}}$ and $f_{\text{interact}}$ are instantiated as two-layer MLPs.

Much prior work has focused on learning to model physical interactions as an end goal. In contrast, we rely on physics predictions only insofar as they affect action planning. To that end, it is more important to know the resultant effects of an action than to make predictions at a fixed time interval. We therefore only need to make a single prediction, $\bar{\mathbf{O}} = f_{\text{physics}}(\mathbf{O})$, to estimate the steady-state configuration of objects as a result of simulating physics indefinitely. This simplification avoids the complications of long-horizon sequential prediction while retaining the information relevant to planning under physical laws and constraints.

## 2.3 RENDERING ENGINE

Because our only supervision occurs at the pixel level, to train our model we learn to map all object-vector predictions back to images. A challenge here lies in designing a function which constructs a single image from an entire collection of objects. The learned renderer consists of two networks, both instantiated as convolutional decoders. The first network predicts an image independently for each input object vector. Composing these images into a single reconstruction amounts to selecting which object is visible at every pixel location. In a traditional graphics engine, this would be accomplished by calculating a depth pass at each location and rendering the nearest object.

To incorporate this structure into our learned renderer, we use the second decoder network to produce a single-channel heatmap for each object. The composite scene image is a weighted average of all of the object-specific renderings, where the weights come from the negative of the predicted heatmaps. In effect, objects with lower heatmap predictions at a given pixel location will be more visible than objects with higher heatmap values. This encourages lower heatmap values for nearer objects. Although this structure is reminiscent of a depth pass in a traditional renderer, the comparison should not be taken literally; the model is only supervised by composite images and no true depth maps are provided during training.

## 2.4 LEARNING OBJECT REPRESENTATIONS

We train the perception, physics, and rendering modules jointly on an image reconstruction and prediction task. Our training data consists of image pairs $(I_0, I_1)$ depicting a collection of objects on a platform before and after a new object has been dropped. ($I_0$ shows one object mid-air, as if being held in place before being released. We refer to Section 3 for details about the generation of training data.) We assume access to a segmentation $\mathbf{S}_0$ for the initial image $I_0$.

Given the observed segmented image $\mathbf{S}_0$, we predict object representations using the perception module $\mathbf{O} = f_{\text{percept}}(\mathbf{S_0})$ and their time-evolution using the physics module $\bar{\mathbf{O}} = f_{\text{physics}}(\mathbf{O})$. The rendering engine then predicts an image from each of the object representations: $\hat{I}_0 = f_{\text{render}}(\mathbf{O}), \hat{I}_1 = f_{\text{render}}(\bar{\mathbf{O}})$.

We compare each image prediction $\hat{I}_t$ to its ground-truth counterpart using both $\mathcal{L}_2$ distance and a perceptual loss $\mathcal{L}_{\text{VGG}}$. As in Johnson et al. (2016), we use $\mathcal{L}_2$ distance in the feature space of a

---

**Algorithm 1** Planning Procedure

    **Input** perception, physics, and rendering modules $f_{\text{percept}}, f_{\text{physics}}, f_{\text{render}}$
    **Input** goal image $I^{\text{goal}}$ with $N$ segments $\mathbf{S}^{\text{goal}} = \{s_k^{\text{goal}}\}_{k=1\ldots N}$
 1: Encode the goal image into a set of $N$ object representations $\mathbf{O}^{\text{goal}} = \{o_k^{\text{goal}}\}_{k=1\ldots N} = f_{\text{percept}}(\mathbf{S}^{\text{goal}})$
 2: **while** $\mathbf{O}^{\text{goal}}$ is nonempty **do**
 3:     Segment the objects that have already been placed to yield $\mathbf{S}^{\text{curr}}$
 4:     **for** $m = 1$ to $M$ **do**
 5:         Sample action $a_m$ of the form (*shape, position, orientation, color*) from uniform distribution
 6:         Observe action $a_m$ as a segment $s_m$ by moving object to specified position and orientation
 7:         Concatenate the observation and segments of existing objects $\mathbf{S}^m = \{s_m\} \cup \mathbf{S}^{\text{curr}}$
 8:         Encode segments $\mathbf{S}^m$ into a set of object representations $\mathbf{O}^m = f_{\text{percept}}(\mathbf{S}^m)$
 9:         Predict the effects of simulating physics on the object representations $\bar{\mathbf{O}}^m = f_{\text{physics}}(\mathbf{O}^m)$
10:         Select the representation $\bar{o} \in \bar{\mathbf{O}}^m$ of the object placed by sampled action $a_m$
11:         Find the goal object $g_m$ that is closest to $\bar{o}$: $g_m = \arg\min_i ||o_i^{\text{goal}} - \bar{o}||_2$
12:         Compute the corresponding distance $d_m = ||o_{g_m}^{\text{targ}} - \bar{o}||_2$
13:     **end for**
14:     Select action $a_{m^*}$ with the minimal distance to its nearest goal object: $m^* = \arg\min_m d_m$.
15:     Execute action $a_{m^*}$ and remove object $g_{m^*}$ from the goal $\mathbf{O}^{\text{goal}} = \mathbf{O}^{\text{goal}} \backslash \{o_{g_{m^*}}^{\text{goal}}\}$.
16: **end while**

---

pretrained VGG network (Simonyan & Zisserman, 2014) as a perceptual loss function. The perception module is supervised by the reconstruction of $I_0$, the physics engine is supervised by the reconstruction of $I_1$, and the rendering engine is supervised by the reconstruction of both images. Specifically, $\mathcal{L}_{\text{percept}}(\cdot) = \mathcal{L}_2(\hat{I}_0, I_0) + \mathcal{L}_{\text{VGG}}(\hat{I}_0, I_0)$, $\mathcal{L}_{\text{physics}}(\cdot) = \mathcal{L}_2(\hat{I}_1, I_1) + \mathcal{L}_{\text{VGG}}(\hat{I}_1, I_1)$, and $\mathcal{L}_{\text{render}}(\cdot) = \mathcal{L}_{\text{percept}}(\cdot) + \mathcal{L}_{\text{physics}}(\cdot)$.

## 2.5 Planning with Learned Models

We now describe the use of our perception, physics, and rendering modules in the representative planning task depicted in Figure 1, in which the goal is to build a block tower to match an observed image. Here, matching a tower does not refer simply to producing an image from the rendering engine that looks like the observation. Instead, we consider the scenario where the model must output a sequence of actions to construct the configuration.

This setting is much more challenging because there is an implicit sequential ordering to building such a tower. For example, the bottom cubes must be placed before the topmost triangle. O2P2 was trained solely on a pixel-prediction task, in which it was never shown such valid action orderings (or any actions at all). However, these orderings are essentially constraints on the physical stability of intermediate towers, and should be derivable from a model with sufficient understanding of physical interactions.

Although we train a rendering function as part of our model, we guide the planning procedure for constructing towers solely through errors in the learned object representation space. The planning procedure, described in detail in Algorithm 1, can be described at a high level in four components:

1. The perception module **encodes** the segmented goal image into a set of object representations $\mathbf{O}^{\text{goal}}$.

2. We **sample** actions of the form *(shape, position, orientation, color)*, where *shape* is categorical and describes the type of block, and the remainder of the action space is continuous and describes the block's appearance and where it should be dropped.

3. We **evaluate** the samples by likewise encoding them as object vectors and comparing them with $\mathbf{O}^{\text{goal}}$. We view action sample $a_m$ as an image segment $s_m$ (analogous to observing a block held in place before dropping it) and use the perception module to produce object vectors $\mathbf{O}^m$. Because the actions selected should produce a stable tower, we run these object representations through the physics engine to yield $\bar{\mathbf{O}}^m$ before comparing with $\mathbf{O}^{\text{goal}}$. The cost is the $\mathcal{L}_2$ distance between the object $\bar{\mathbf{o}} \in \bar{\mathbf{O}}^m$ corresponding to the most recent action and the goal object in $\mathbf{O}^{\text{goal}}$ that minimizes this distance.

4. Using the action sampler and evaluation metric, we **select** the sampled action that minimizes $\mathcal{L}_2$ distance. We then **execute** that action in MuJoCo (Todorov et al., 2012). We continue this procedure, iteratively re-planning and executing actions, until there are as many actions in the

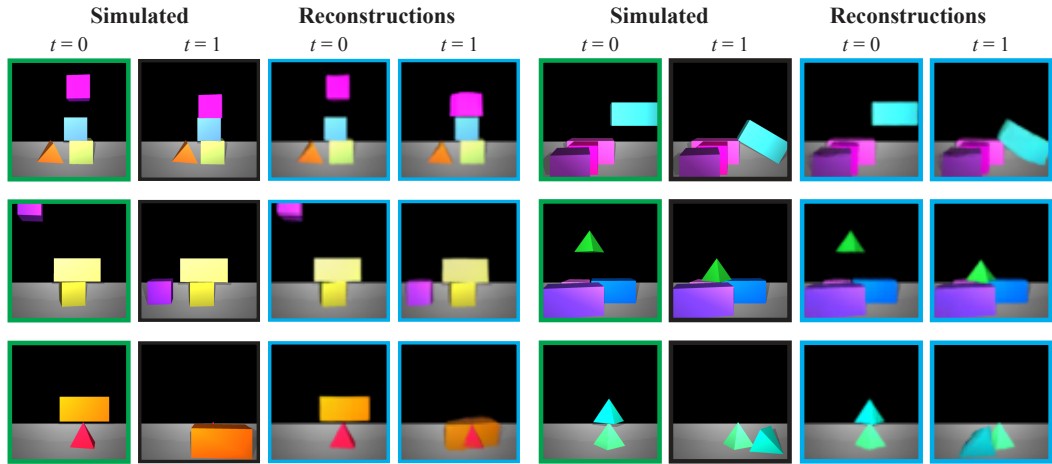

Figure 3: Given an observed segmented image $I_0$ at $t = 0$, our model predicts a set of object representations $\mathbf{O}$, simulates the objects with a learned physics engine to produce $\bar{\mathbf{O}} = f_{\text{physics}}(\mathbf{O})$, and renders the resulting predictions $\hat{I} = f_{\text{render}}(\bar{\mathbf{O}})$, the scene's appearance at a later time. We use the convention (in all figures) that observations are outlined in **green**, other images rendered with the ground-truth renderer are outlined in **black**, and images rendered with our learned renderer are outlined in **blue**.

executed sequence as there are objects in the goal image. In the simplest case, the distribution from which actions are sampled may be uniform, as in Algorithm 1. Alternatively, the cross-entropy method (CEM) (Rubinstein & Kroese, 2004) may be used, repeating the sampling loop multiple times and fitting a Gaussian distribution to the lowest-cost samples. In practice, we used CEM starting from a uniform distribution with five iterations, 1000 samples per iteration, and used the top 10% of samples to fit the subsequent iteration's sampling distribution.

## 3 EXPERIMENTAL EVALUATION

In our experimental evaluation, we aim to answer the following questions, (1) After training solely on physics prediction tasks, can O2P2 reason about physical interactions in an actionable and useful way? (2) Does the implicit object factorization imposed by O2P2's structure provide a benefit over an object-agnostic black-box video prediction approach? (3) Is an object factorization still useful even without supervision for object representations?

### 3.1 IMAGE RECONSTRUCTION AND PREDICTION

We trained O2P2 to reconstruct observed objects and predict their configuration after simulating physics, as described in Section 2.4. To generate training data, we simulated dropping a block on top of a platform containing up to four other blocks. We varied the position, color, and orientation of three block varieties (cubes, rectangular cuboids, and triangles). In total, we collected 60,000 training images using the MuJoCo simulator. Since our physics engine did not make predictions at every timestep (Section 2.2), we only recorded the initial and final frame of a simulation. For this synthetic data, we used ground truth segmentations corresponding to visible portions of objects.

Representative predictions of our model for image reconstruction (without physics) and prediction (with physics) on held-out random configurations are shown in Figure 3. Even when the model's predictions differed from the ground truth image, such as in the last row of the figure, the physics engine produced a plausible steady-state configuration of the observed scene.

### 3.2 BUILDING TOWERS

After training O2P2 on the random configurations of blocks, we fixed its parameters and employed the planning procedure as described in Section 2.5 to build tower configurations observed in images. We also evaluated the following models as comparisons:

- **No physics** is an ablation of our model that does not run the learned physics engine, but instead simply sets $\bar{\mathbf{O}} = \mathbf{O}$
- Stochastic adversarial video prediction (**SAVP**), a block-box video prediction model which does not employ an object factorization Lee et al. (2018). The cost function of samples is evaluated directly on pixels. The sampling-based planning routine is otherwise the same as in ours.

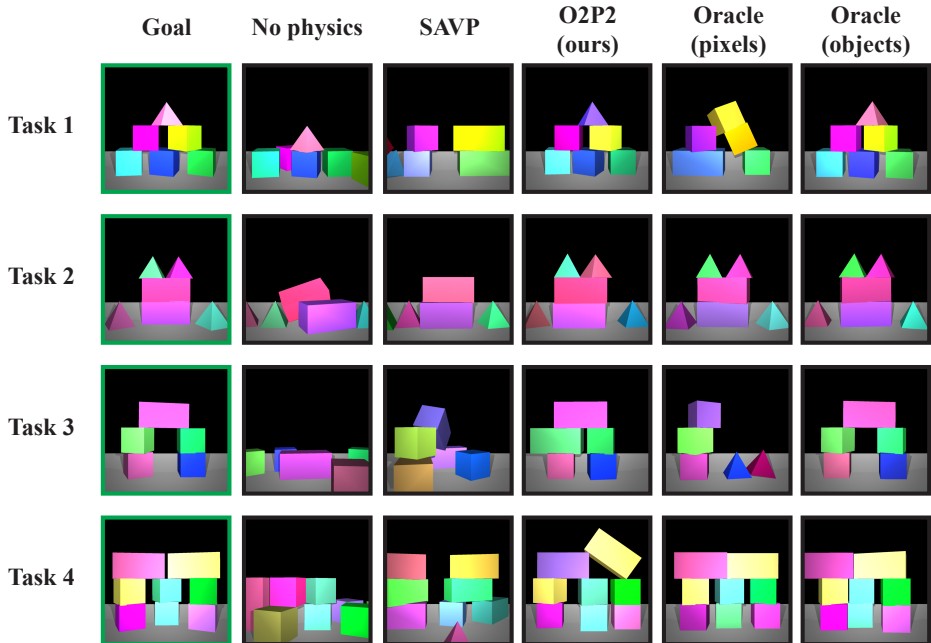

Figure 4: Qualitative results on building towers using planning. Given an image of the goal tower, we can use the learned object representations and predictive model in O2P2 for guiding a planner to place blocks in the world and recreate the configuration. We compare with an ablation, an object-agnostic video prediction model, and two 'oracles' with access to the ground-truth simulator.

Table 1: Accuracy (%) of block tower builds by our approach and the four comparison models. Our model outperforms Oracle (pixels) despite not having the ground-truth simulator by virtue of a more appropriate object-factorized objective to guide the planning procedure.

| No physics | SAVP | Ours | Oracle (pixels) | Oracle (objects) |
|---|---|---|---|---|
| 0 | 24 | 76 | 71 | 92 |

- **Oracle (pixels)** uses the MuJoCo simulator to evaluate samples instead of our learned physics and graphics engines. The cost of a block configuration is evaluated directly in pixel space using $\mathcal{L}_2$ distance.
- **Oracle (objects)** also uses MuJoCo, but has access to segmentation masks on input images while evaluating the cost of proposals. Constraining proposed actions to account for only a single object in the observation resolves some of the inherent difficulties of using pixel-wise loss functions.

Qualitative results of all models are shown in Figure 4 and a quantitative evaluation is shown in Table 1. We evaluated tower stacking success by greedily matching the built configuration to the ground-truth state of the goal tower, and comparing the maximum object error (defined on its position, identity, and color) to a predetermined threshold. Although the threshold is arbitrary in the sense that it can be chosen low enough such that all builds are incorrect, the relative ordering of the models is robust to changes in this value. All objects must be of the correct shape for a built tower to be considered correct, meaning that our third row prediction in Figure 4 was incorrect because a green cube was mistaken for a green rectangular cuboid.

While SAVP made accurate predictions on the training data, it did not generalize well to these more complicated configurations with more objects per frame. As such, its stacking success was low. Physics simulation was crucial to our model, as our No-physics ablation failed to stack any towers correctly. We explored the role of physics simulation in the stacking task in Section 3.3. The 'oracle' model with access to the ground-truth physics simulator was hampered when making comparisons in pixel space. A common failure mode of this model was to drop a single large block on the first step to cover the visual area of multiple smaller blocks in the goal image. This scenario was depicted by the blue rectangular cuboid in the first row of Figure 4 in the Oracle (pixels) column.

## 3.3 THE IMPORTANCE OF UNDERSTANDING PHYSICS

Figure 5 depicts the entire planning and execution procedure for O2P2 on a pyramid of six blocks. At each step, we visualize the process by which our model selects an action by showing a heatmap of

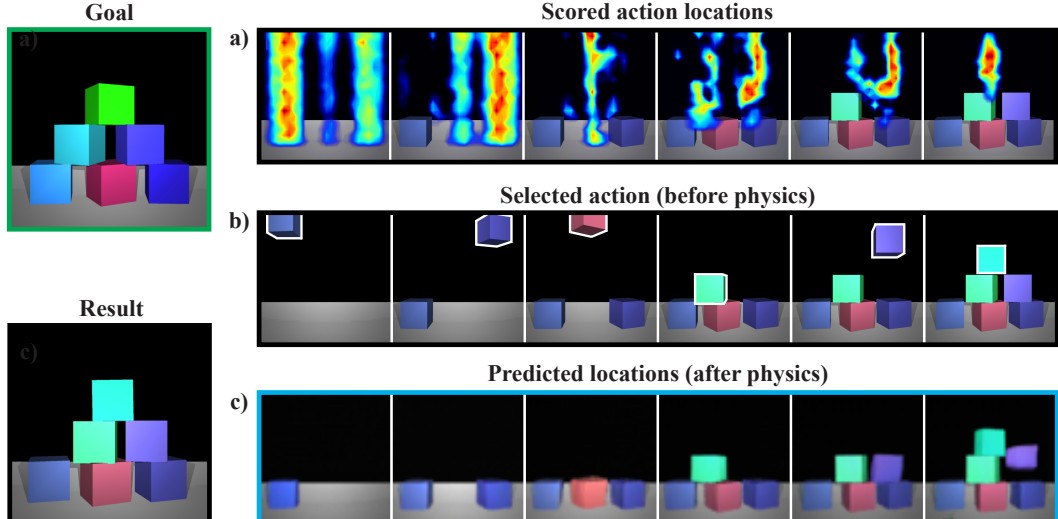

Figure 5: (a) Visualization of scored locations for dropping an object at each timestep. Because O2P2 simulates physics before selecting an action, it is able to plan a sequence of stable actions. (b) The selected block and drop position from the scored samples, outlined in white. (c) The prediction from our physics model of the result of running physics on the selected block.

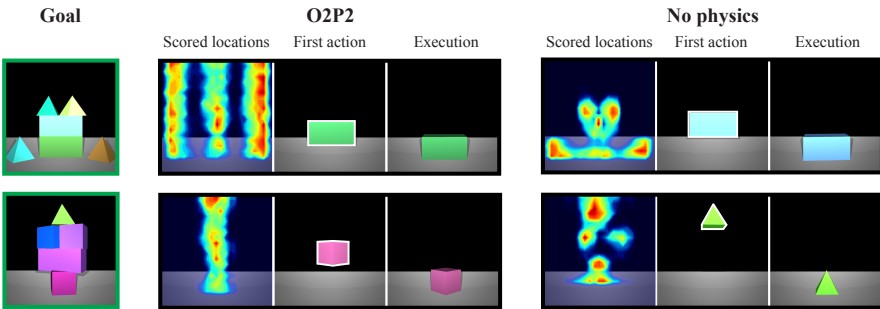

Figure 6: Heatmaps showing sampled action scores for the initial action given a goal block tower. O2P2's scores reflect that the objects resting directly on the platform must be dropped first, and that they may be dropped from any height because they will fall to the ground. The No-physics ablation, on the other hand, does not implicitly represent that the blocks need to be dropped in a stable sequence of actions because it does not predict the blocks moving after being released.

scores (negative MSE) for each action sample according to the sample's $(x, y)$ position (Figure 5a). Although the model is never trained to produce valid action decisions, the planning procedure selects a physically stable sequence of actions. For example, at the first timestep, the model scores three $x$-locations highly, corresponding to the three blocks at the bottom of the pyramid. It correctly determines that the height at which it releases a block at any of these locations does not particularly matter, since the block will drop to the correct height after running the physics engine. Figure 5b shows the selected action at each step, and Figure 5c shows the model's predictions about the configuration after releasing the sampled block.

Similar heatmaps of scored samples are shown for the No-physics ablation of our model in Figure 6. Because this ablation does not simulate the effect of dropping a block, its highly-scored action samples correspond almost exactly to the actual locations of the objects in the goal image. Further, without physics simulation it does not implicitly select for stable action sequences; there is nothing to prevent the model from selecting the topmost block of the tower as the first action.

**Planning for alternate goals.** By implicitly learning the underlying physics of a domain, our model can be used for various tasks besides matching towers. In Figure 7a, we show our model's representations being used to plan a sequence of actions to maximize the height of a tower. There is no observation for this task, and the action scores are calculated based on the highest non-zero pixels after rendering samples with the learned renderer. In Figure 7b, we consider a similar sampling procedure as in the tower-matching experiments, except here only a single *unstable* block is shown. Matching a free-floating block requires planning with O2P2 for multiple steps at once.

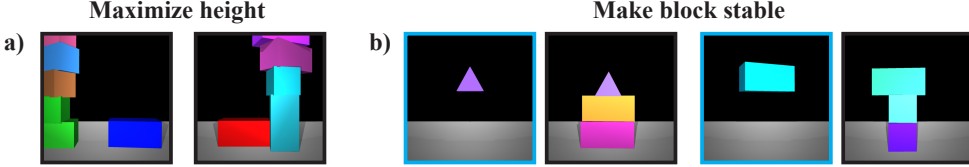

Figure 7: O2P2 being used to plan for the alternate goals of (a) maximizing the height of a tower and (b) making an observed block stable by use of any other blocks.

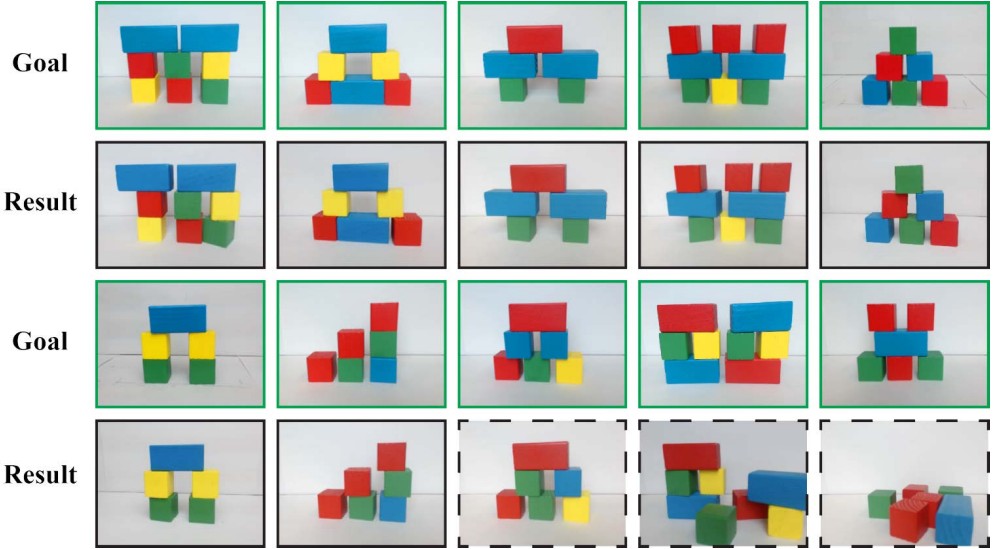

Figure 8: Ten goal images alongside the result of the Sawyer's executed action sequence using O2P2 for planning. The seven action sequences counted as correct are outlined in solid black; the three counted as incorrect are outlined in dashed lines. We refer the reader to Appendix B for more evaluation examples and people.eecs.berkeley.edu/~janner/o2p2 for videos of the evaluation.

## 3.4 TRANSFER TO ROBOTIC ARM

We evaluated O2P2 on a Sawyer robotic arm using real image inputs. We deployed the same perception, physics, and rendering modules used on synthetic data with minor changes to the planning procedure to make real-world evaluation tractable. Instead of evaluating a sampled action by moving an appropriate block to the specified position and inferring object representations with the perception module, we trained a separate two-layer MLP to map directly from actions to object representations. We refer to this module as the embedder: $o_m = f_{\text{embedder}}(a_m)$.

Mapping actions to object representations removed the need to manually move every sampled block in front of the camera, which would have been prohibitively slow on a real robot. The embedder was supervised by the predicted object representations of the perception module on real image inputs; we collected a small dataset of the Sawyer gripper holding each object at one hundred positions and recorded the ground truth position of the gripper along with the output of the perception module for the current observation.

The embedder took the place of lines 6-8 of Algorithm 1. We also augmented the objective used to select actions in line 11. In addition to $\mathcal{L}_2$ distance between goal and sampled object representations, we used a pixelwise $\mathcal{L}_2$ distance between the observed and rendered object segments and between the rendered object segments before and after use of the physics module. The latter loss is useful in a real setting because the physical interactions are less predictable than their simulated counterparts, so by penalizing any predicted movement we preferentially placed blocks directly in a stable position.

By using end-effector position control on the Sawyer gripper, we could retain the same action space as in synthetic experiments. Because the *position* component of the sampled actions referred to the block placement location, we automated the picking motion to select the sampled block based on the *shape* and *color* components of an action. Real-world evaluation used colored wooden cubes and rectangular cuboids.

Real image object segments were estimated by applying a simple color filter and finding connected components of sufficient size. To account for shading and specularity differences, we replaced all pixels within an object segment by the average color within the segment. To account for noisy segment masks, we replaced each mask with its nearest neighbor (in terms of pixel MSE) in our MuJoCo-rendered training set.

We tested O2P2 on twenty-five goal configurations total, of which our model correctly built seventeen. Ten goal images, along with the result of our model's executed action sequence, are shown in Figure 8. The remainder of the configurations are included in Appendix B.

## 4    RELATED WORK

Our work is situated at the intersection of two distinct paradigms. In the first, a rigid notion of object representation is enforced via supervision of object properties (such as size, position, and identity). In the second, scene representations are not factorized at all, so no extra supervision is required. These two approaches have been explored in a variety of domains.

**Image and video understanding.**    The insight that static observations are physically stable configurations of objects has been leveraged to improve 3D scene understanding algorithms. For example, Zheng et al. (2014); Gupta et al. (2010); Shao et al. (2014); Jia et al. (2015) build physically-plausible scene representations using such stability constraints. We consider a scenario in which the physical representations are learned from data instead of taking on a predetermined form. Wu et al. (2017b;a) encode scenes in a markup-style representation suitable for consumption by off-the-shelf rendering engines and physics simulators. In contrast, we do not assume access to supervision of object properties (only object segments) for training a perception module to map into a markup language.

There has also been much attention on inferring object-factorized, or otherwise disentangled, representations of images (Eslami et al., 2016; Greff et al., 2017; van Steenkiste et al., 2018). In contrast to works which aim to discover objects in a completely unsupervised manner, we focus on using object representations learned with minimal supervision, in the form of segmentation masks, for downstream tasks. Object-centric scene decompositions have also been considered as a potential state representation in reinforcement learning (Diuk et al., 2008; Scholz et al., 2014; Devin et al., 2017; Goel et al., 2018; Keramati et al., 2018). We are specifically concerned with the problem of predicting and reasoning about physical phenomena, and show that a model capable of this can also be employed for decision making.

**Learning and inferring physics.**    Fragkiadaki et al. (2016); Watters et al. (2017); Chang et al. (2016) have shown approaches to learning a physical interaction engine from data. Hamrick et al. (2011) use a traditional physics engine, performing inference over object parameters, and show that such a model can account for humans' physical understanding judgments. We consider a similar physics formulation, whereby update rules are composed of sums of pairwise object-interaction functions, and incorporate it into a training routine that does not have access to ground truth supervision in the form of object parameters (such as position or velocity).

An alternative to using a traditional physics engine (or a learned object-factorized function trained to approximate one) is to treat physics prediction as an image-to-image translation or classification problem. In contrast to these prior methods, we consider not only the accuracy of the predictions of our model, but also its utility for downstream tasks that are intentionally constructed to evaluate its ability to acquire an actionable representation of intuitive physics. Comparing with representative video prediction (Lee et al., 2018; Babaeizadeh et al., 2018) and physical prediction (Ehrhardt et al., 2017; Mottaghi et al., 2016; Li et al., 2017; Lerer et al., 2016) methods, our approach achieves substantially better results at tasks that require building structures out of blocks.

## 5    CONCLUSION

We introduced a method of learning object-centric representations suitable for physical interactions. These representations did not assume the usual supervision of object properties in the form of position, orientation, velocity, or shape labels. Instead, we relied only on segment proposals and a factorized structure in a learned physics engine to guide the training of such representations. We demonstrated that this approach is appropriate for a standard physics prediction task. More importantly, we showed that this method gives rise to object representations that can be used for difficult planning problems, in which object configurations differ from those seen during training, without further adaptation. We evaluated our model on a block tower matching task and found that it outperformed object-agnostic approaches that made comparisons in pixel-space directly.

ACKNOWLEDGMENTS

We thank Michael Chang for insightful discussion and anonymous reviewers for feedback on an early draft of this paper. This work was supported by the National Science Foundation Graduation Research Fellowship and the Open Philanthropy Project AI Fellowship.

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

## A  IMPLEMENTATIONS DETAILS

Objects were represented as 256-dimensional vectors. The perception module had four convolutional layers of {32, 64, 128, 256} channels, a kernel size of 4, and a stride of 2 followed by a single fully-connected layer with output size matching the object representation dimension. Both MLPs in the physics engine had two hidden layers each of size 512. The rendering networks had convolutional layers with {128, 64, 32, 3} channels (or 1 output channel in the case of the heatmap predictor), kernel sizes of {5, 5, 6, 6}, and strides of 2. We used the Adam optimizer (Kingma & Ba, 2015) with a learning rate of 1e-3.

## B  SAWYER RESULTS

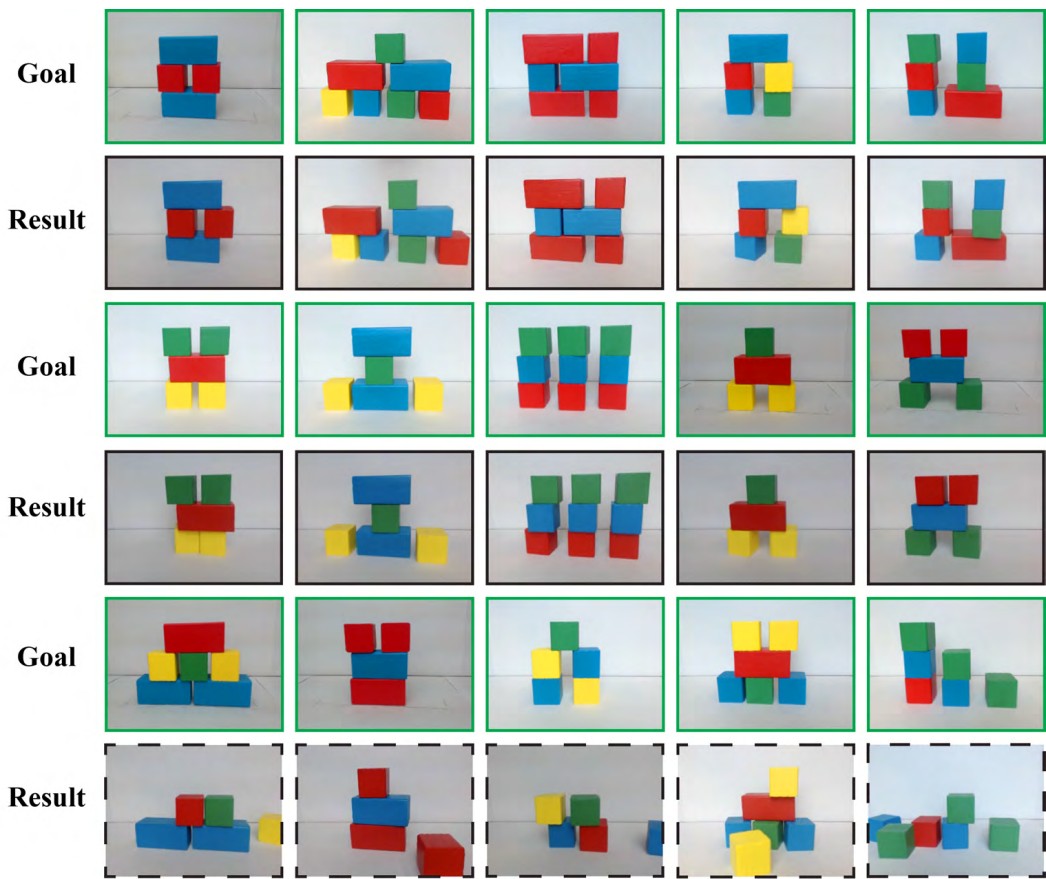

Figure 9: Extension of Figure 8, showing our planning results on a Sawyer arm with real image inputs. The seven action sequences counted as correct are outlined in solid black; the three counted as incorrect are outlined in dashed lines.

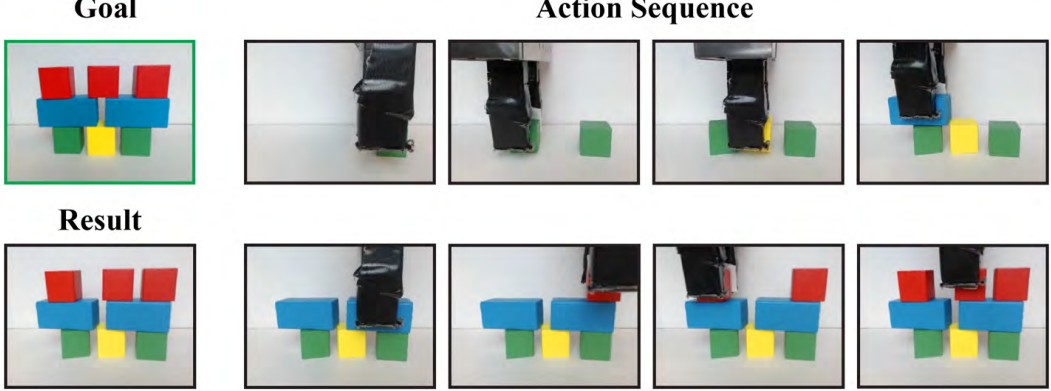

Figure 10: All actions taken by our planning procedure for one of the goal configurations from Figure 8.

