# OpenReview forum: " Reasoning About Physical Interactions with Object-Oriented Prediction and Planning"
_ICLR.cc/2019/Conference_

### Official Review · AnonReviewer3 · 2018-10-30
**Interesting paper; some assumptions should be stated more clearly**

**Rating:** 9
**Confidence:** 4

**Review:**

edit:  the authors nicely revised the submission, I think it is a very good paper. I increased my rating.

-----

This paper presents a method that learns to reproduce 'block towers' from a given image. A perception model, a physics engine model, and a rendering engine are first trained together on pairs of images.
The perception model predicts a representation of the scene decomposed into objects;  the physics engine predicts the object representation of a scene from an initial object representation; the rendering engine predicts an image given an object representation.

Each training pair of images is made of the first image of a sequence when introducing an object into a scene, and of the last image of the sequence, after simulating the object's motion with a physics engine. The 3 parts of the pipeline (perception, physics, rendering) are trained together on this data.

To validate the learned pipeline, it is used to recreate scenes from reference images, by trying to introduce objects in an empty scene until the given scene can be reproduced. It outperforms a related pipeline that lacks a scene representation based on objects.

This is a very interesting paper, with new ideas:
- The object-based scene representation makes a lot of sense, compared to the abstract representation used in recent work.
- The training procedure, based on observing the result of an action, is interesting as the examples are easy to collect (except for the fact that the ground truth segmentation of the images is used as input, see below).

However, there are several things that are swept 'under the carpet' in my opinion, and this should be fixed if the paper is accepted.

* the input images are given in the form of a set of images, one image corresponding to the object segmentation. This is mentioned only once (briefly) in the middle of the paragraph for Section 2.1, while this should be mentioned in the introduction, as this makes the perception part easier. There is actually a comment in the discussion section and the authors promised to clarify this aspect, which should indeed be more detailed. For example, do the segments correspond to the full objects, or only the visible parts?

* The training procedure is explained only in Section 4.1. Before reaching this part, the method remained very mysterious to me. The text in Section 4.1 should be moved much earlier in the paper, probably between current sections 2.3 and 2.4, and briefly explained in the introduction as well.
This training procedure is in fact fully supervised - which is fine with me: Supervision makes learning 'safer'. What is nice here is that the training examples can be collected easily - even if the system was not running in a simulation.

* if I understand correctly the planning procedure, it proceeds as follows:
- sampling 'actions' that introduce 1 object at a time (?)
- for each sampled action, predicting the scene representation after the action is performed, by simulating it with the learned pipeline,
- keeping the action that generates a scene representation close to the scene representation computed for the goal image of the scene.
- performing the selected action in a simulator, and iterate until the number of performed actions is the same as the number of objects (which is assumed to be known).

-> how do you compare the scene representation of the goal image and the predicted one before the scene is complete? Don't you need some robust distance instead of the MSE?
-> are the actions really sampled randomly?  How many actions do you need to sample for the examples given in the paper?

I also have one question about the rendering engine:  Why using the weighted average of the object images? Why not using the intensity of the object with the smallest predicted depth?  It should generate sharper images. Does using the weighted average make the convergence easier?

---

> ### Author Response · Authors · 2018-11-19
> **Response to Reviewer 3**
>
> Thank you for your thorough feedback. We have uploaded a revision to make the model and planning procedure clearer. We will upload a second revision this coming week to include results on a Sawyer robot using real image inputs (qualitative results are given below in a video link).
>
> -- Use of segments in lieu of object property supervision
> We have made more explicit the use of object segmentation in the Section 2 notation (where we describe the model and planning procedure). The segmentations correspond to only the visible parts of objects. We have clarified this in Section 3.1, where we describe data collection.
>
> We have evaluated our approach on a Sawyer arm using physical blocks to demonstrate applicability to the real world (goo.gl/151BT1). Here we used simple color cues to segment the image observations.
>
> -- Planning procedure
> We have added an algorithmic description on page 4 (Algorithm 1: Planning Procedure) to make this section clearer.  To answer your question about comparing scenes with different number of objects: we match a proposed object to the goal object which minimizes L2 distance in the learned object representations. Some goal objects will be unaccounted for until the last step of the planning algorithm, when there is an action for each object in the goal image.
>
> We have added details of the cross-entropy method (CEM) to Step 4 of Section 2.5. We sampled actions beginning from a uniform distribution and used CEM to update the sampling distribution. We used 5 CEM iterations with 1000 samples per iteration. Because all of the samples could be evaluated in batch-mode, there was little overhead to evaluating a large number of samples.
>
> -- Training procedure
> Per your suggestion, we have moved the training procedure to come after the model description and before the planning algorithm.
>
> -- Clarification on rendering module
> We use a weighted average for composing individual object images so that the rendering process is fully differentiable. This design decision makes end-to-end training of the perception, physics, and rendering modules easier.

---

### Official Review · AnonReviewer1 · 2018-10-31
**Intriguing idea, but the paper is not sufficiently clear, and the experimental evaluation is weak.**

**Rating:** 7
**Confidence:** 4

**Review:**

Summary:
The paper presents a platform for predicting images of objects interacting with each other under the effect of gravitational forces. Given an image describing the initial arrangement of the objects in a scene, the proposed architecture first detects the objects and encode them using a perception module. A physics module then predicts the final arrangement of the object after moving under the effects of gravity. A rendering module takes as input the predicted final positions of objects and returns an image. The proposed architecture is trained by using pixel labels only, by reducing the gaps between the predicted rendered images and the images returned by the MuJuCo physics engine. This error's gradient is back-propagated to the physics and perception modules. The proposed platform is also used for planning object placements by sampling a large number of object shapes, orientations and colors, predicting the final configurations, and selecting initial placements that lead to final configurations that are as close as possible to given goal configurations using the L2 norm in the VGG features. Experiments performed in a simple blocks world show that the proposed approach is not only useful for prediction, but can also be used for planning object placements.
Clarity:
The paper is not very well written. The description of the architecture should be much more precise. Some details are given right before the conclusion, but they are still just numbers and leave a lot of questions unanswered. For instance, the perception module is explained in only a few line in subsection 2.1. Some concrete examples could help here. How are the object proposals defined? How are the objects encoded? What exactly is being encoded here? Is it the position and orientation?
Originality:
The proposed architecture seems novel, but there are many closely related works that are based on the same idea of decomposing the system into a perception,  a physics simulation, and a rendering module. Just from the top of my head, I can think of the SE3-Nets. There is also a large body of work from the group of Josh Tanenbaum on similar problems of learning physics and rendering. I think this concept is not novel anymore and the expectations should be raised to real applications.
Significance:
The simplicity of the training process that is fully based on pixel labeling makes this work interesting. There are however some issues related to the experimental evaluation that remains unsatisfactory. First, all the experiments are performed on a single benchmark, we cannot easily draw conclusions about a given algorithm based on a single benchmark. Second, this is a toy benchmark that with physical interactions that are way less complex than interactions that happen between real objects. The objects are also not diverse enough in their appearances and textures. I wonder why the authors avoided collecting a dataset of real images of objects and using it to evaluate their algorithm instead of the toy artificial data. I also suspect that with 60k training images, you can easily overfit this task. How can this work generalize to real physical interactions? How can you capture mass and friction, for example?
Planning is based on sampling objects of different shapes and colors, do you assume the existence of such library in advance?
The baselines that are compared to are also not very appropriate. For instance, comparing to no physics does not add much information. We know that the objects will fall after they are dropped, so the "no physics" baseline will certainly perform badly. Comparisons to SAVP are also unfair because it requires previous frames, which are not provided here, and SAVP is typically used for predicting the very next frames and not the final arrangements of objects, as done here.
In summary: I think the authors are on something here and the idea is great. However, the paper needs to be made much clearer and more precise, and the experimental evaluation should be improved by performing experiments in a real-world environment. Otherwise, this paper will not have much impact.

Post-rebuttal update:
The paper was substantially improved. New experiments using real objects have been included, this clearly demonstrates the merits of the proposed method in robotic object manipulation.

---

> ### Author Response · Authors · 2018-11-19
> **Response to Reviewer 1**
>
> Thank you for your thorough feedback. To address your comment about experiments in a real-world environment, we have tested our model on a Sawyer robot with real camera images. A representative video can be found here:
> goo.gl/151BT1
> We will update the paper to include these results this coming week. Additionally, we have already updated the paper to make the model and planning procedure clearer. Below, we describe some of these changes.
>
> -- Clarification on object encodings
> We have explained more thoroughly that the object encodings are not supervised directly to have semantically meaningful components like position or orientation. As compared to most prior work on object-factorized representations, we do not assume access to ground truth properties for the objects. This is why the perception module cannot be trained independently; we have no supervision for its outputs. Instead, we train the perception, graphics, and physics modules jointly to reconstruct the current observations and predict the subsequent observation (Figure 2c). In this way, the object representations come to encode these attributes without direct supervision of such properties. Of course, learning representations via a reconstruction objective is not unique to our paper; what we show is that these representations can be sufficient for planning in physical understanding tasks.
>
> -- Relation to prior work
> The most relevant works about learning physics and rendering you might be referring to are Neural Scene De-rendering (NSD) and Visual Scene De-animation (VDA). These works learn object encodings by direct supervision of properties like position and orientation. As discussed in the previous section, we weaken the requirement for ground-truth object properties, instead requiring only segments instead of attribute annotations. We previously cited VDA and have now added NSD along with a short description of this supervision difference.
>
> SE3-Nets used point cloud shape representations, whereas we use learned representations driven by an image prediction task.
>
> -- Generalization to real physical interactions
> We have now demonstrated our model in the physical world (see video link given above).
>
> -- No-physics baseline, SAVP
> Yes, it is not surprising that a model which did not predict physics did not perform well. We included this model as an ablation because we can better understand how our full model makes decisions by comparing it to the physics-ablated version, as in Figure 6. The SAVP baseline takes in a previous frame in the form of an object sample, similar to how our model views a sample by rendering an object mid-air, allowing for a head-to-head comparison to a black-box frame prediction approach.

---

### Official Review · AnonReviewer2 · 2018-11-03
**Very good idea, lacks in presentation/formalization and in experimental evaluation**

**Rating:** 5
**Confidence:** 5

**Review:**

A method is proposed, which learns to reason on physical interactions of different objects (solids like cuboids, tetrahedrons etc.). Traditionally in related work the goal is to predict/forecast future observations, correctly predicting (and thus learning) physics. This is also the case in this paper, but the authors explicitly state that the target is to evaluate the learned model on downstream tasks requiring a physical understanding of the modelled environment.

The main contribution here lies in the fact that no supervision is used for object properties. Instead, a mask predictor is trained without supervision, directly connected to the rest of the model, ie. to the physics predictor and the output renderer. The method involves a planning phase, were different objects are dropped on the scene in the right order, targeting bottom objects first and top objects later. The premise here is that predicting the right order of the planning actions requires understanding the physics of the underlying scene.

I particularly appreciated the fact, that object instance renderers are combined with a global renderer, which puts individual images together using predicted heatmaps for each object. With a particular parametrization, these heatmaps could be related to depth maps allowing correct depth ordering, but depth information has not been explicitly provided during training.

Important issues:

One of the biggest concerns is the presentation of the planning algorithm, and more importantly, a proper formalization of what is calculated, and thus a proper justification of this part. The whole algorithm is very vaguely described in a series of 4 items on page 4. It is intuitively almost clear how these steps are performed, but the exact details are vague. At several steps, calculated entities are “compared” to other entities, but it is never said what this comparison really results in. The procedure is reminiscent of particle filtering, in that states (here: actions) are sampled from a distribution and then evaluated through a likelihood function, resulting in resampling. However, whereas in particle filtering there is clear probabilistic formalization of all key quantities, in this paper we only have a couple of phrases which describe sampling and “comparisons” in a vague manner.

Since the procedure performs planning by predicting a sequence of actions whose output at the end can be evaluated, thus translated into a reward, I would have also liked a discussion (or at least a remark) why reinforcement learning has not been considered here.

I am also concerned by an overclaim of the paper. As opposed to what the paper states in various places, the authors really only evaluate the model on video prediction and not on other downstream tasks. A single downstream task is very briefly mentioned in the experimental section, but it is only very vaguely described, it is unclear what experiments have been performed and there is no evaluation whatsoever.

Open questions:

Why is the proposed method better than one of the oracles?

Minor remarks:

It is unclear what we see in image 4, as there is only a single image for each case (=row) and method (=column).

The paper is not fully self-contained. Several important aspects are only referred to by citing work, e.g. CEM sampling and perceptual loss. These are concepts which are easy to explain and which do not take much space. They should be added to the paper.

A threshold is mentioned in the evaluation section. A plot should be given showing the criterion as a function of this threshold, as is standard in, for instance, pose estimation literature.

I encourage the authors to use the technical terms “unary terms” and “binary terms” in the equation in section 2.2. This is the way how the community referred to interactions in graphical models for relational reasoning long before deep learning showed up on the horizon, let’s be consistent with the past.

I do not think that the physics module can be reasonable be called a “physics simulator” as has been done throughout the paper. It does not simulate physics, it predicts physics after learning, which is not a simulation.

A cube has not been confused with a rectangle, as mentioned in the paper, but with a rectangular cuboid. A rectangle is a 2D shape, a rectangular cuboid is a 3D polyhedron.

---

> ### Author Response · Authors · 2018-11-19
> **Response to Reviewer 2**
>
> Thank you for your feedback and suggestions. We have updated the paper to make the planning algorithm clearer, give short descriptions of CEM and perceptual losses, and incorporate your terminology suggestions (‘rectangular cuboid’, ‘unary’, ‘binary’, etc). At the request of other reviewers, we have also tested our approach on a physical Sawyer robot. The following video gives a qualitative result analogous to Figure 4:
> goo.gl/151BT1
> These results will be included in the paper in a second revision this week. Below, we give more details about the current changes.
>
> -- Evaluation on downstream tasks
> Downstream task results were in the original submission (all Figures after 3 and Table 1); we have updated the paper to better differentiate between image prediction results in isolation and the use of our model’s predictions in a planning procedure to build towers.
>
> Figure 4 shows qualitative results on this building task, and Table 1 gives quantitative results. Figures 5 and 6 give some analysis of the procedure by which our model selects actions. Figure 7 briefly shows how our model can be adapted to other physics-based tasks: stacking to maximize height, and building a tower to make a particular block stable.
>
> -- Planning algorithm
> We have added a more precise algorithmic description on page 4 to make the tower-building procedure clearer (Algorithm 1: Planning Procedure).
>
> -- Oracle models
> We have added a sentence to the Table 1 caption to explain why O2P2 outperforms Oracle (pixels). The Oracle (pixels) model has access to the true physics simulator which generated the data, but not an object-factorized cost function. Instead, it uses pixel-wise L2 over the entire image (Section 3.2). The top row of Figure 4 is illustrative here: the first action taken by Oracle (pixels) was to drop the blue rectangular cuboid in the bottom left to account for both of the blue cubes in the target. Our model, despite having a worse physics predictor, performs better by virtue of its object factorization.
>
> -- Figure 4 clarification
> We have updated the caption of Figure 4 and changed some text in the graphic. Figure 4 shows qualitative results on the tower building task described above. We show four goal images (outlined in green), and the towers built by each of five methods. This figure has a few utilities:
>     1. It illustrates what our model’s representations capture well for planning and what they do not. For example, most mistakes made by our model concern object colors. This suggests that object positions are more prominently represented by our model’s representations than color.
>     2. It shows why an object-factorization is still useful even if one has access to the “true” physics simulator (as discussed in the previous question).
>     3. It shows that the types of towers being built in the downstream task are not represented in the training set of the perception, graphics, and physics modules (depicted in Figure 3, where we show reconstruction and prediction results). The object-factorized predictions allow our model to generalize out of distribution more effectively than an object-agnostic video prediction model (Table 1).
>
> -- Reinforcement learning baseline
> We have found that a PPO agent works poorly on this task, possibly due to the high dimensionality of the observation space (raw images). We will continue to try to get this baseline to work for the next revision, and would be happy to try out any other RL algorithms that the reviewer might suggest.

---

> > ### Comment · AnonReviewer2 · 2018-11-26
> > **Unchanged**
> >
> > I thank the authors for the changes made to the document, which clarify some of my questions.
> > I still think that the experimental part of the paper is too weak for a publication at ICLR at this point.

---

### Public Comment · (anonymous) · 2018-10-01
**More implementation details**

This is a very nice paper! However, I wish it included some more details of the implementation (perhaps a future revision could include an appendix?) For example, how did you get the region proposals/segmentation for each video frame? What exactly are the equations involved in the reconstruction process?

---

> ### Author Response · Authors · 2018-10-03
> **Implementation details**
>
> Thank you for your questions. We will include an appendix with more implementation details (currently in Section 5) in the next version. In the meantime, here we describe the reconstruction process in more depth.
>
> 1. The perception network has four convolutional layers (32, 64, 128, 256 channels) with ReLU nonlinearities followed by a fully connected layer. It predicts a set of object representations given an image at t=0:
>
>         o_0 = f_percept(I_0)
>
> 2. The physics engine consists of a pairwise interaction MLP and single-object transition MLP, each with two hidden layers. It predicts object representations at the next timestep given an initial configuration:
>
>         o_1 = f_physics(o_0)
>
> (To see f_physics broken down into separate terms for the two MLPs, see Section 2.2)
>
> 3. The rendering engine has two networks, which we will call f_image and f_heatmap. For each object o_{t,i} in a set of objects o_t at timestep t, f_image predicts a three-channel image and f_heatmap predicts a single-channel heatmap. We render each object separately with f_image and then combine these images by a weighted averaging over objects, where the weights come from the negatives of the heatmaps (passed through a nonlinearity). More precisely, denoting the heatmaps at time t for all objects as
>
> 	H_t = softmax( -f_heatmap(o_t) ),
>
> the j^th pixel of the predicted composite image is then:
>
>         \hat{I}_{t,j} = \sum_i f_image( o_{t,i} )_j * H_{t,i,j},
>
> where H_{t,i,j} is the j^th pixel of the heatmap for the i^th object at time t.
>
> Both networks have a single fully-connected layer followed by four deconvolutional layers with ReLU nonlinearities. f_image has (128, 64, 32, 3) channels and f_heatmap has (128, 64, 32, 1) channels. From here on, we will use f_render to describe this entire process:
>
>         \hat{I}_t = f_render(o_t)
>
> The equations here risk making all of this seem more complicated than it really is. The high-level picture is that we need a way to produce a single image from a set of objects, so we render each object separately and then take a weighted average over the individual images in something that could be thought of as a soft depth pass.
>
> 4. Reconstructing an image at the observed timestep then looks a lot like an auto-encoder:
>
> 	\hat{I}_0 = f_render( f_percept(I_0) )
>
> Reconstructing an image at the next timestep uses the physics engine in between:
>
> 	\hat{I}_1 = f_render( f_physics( f_percept(I_0) ) )
>
> These equations are reflected in the loss functions on page 6. (For example, the physics engine is only trained via the loss from reconstructing I_1, since it is not used in reconstructing I_0.) We used ground truth object segments in our experiments, which we discuss in the answer to the question on 10/02/2018.

---

### Public Comment · (anonymous) · 2018-10-02
**Nice paper, just a few comments**

Dear authors, I think conceptually that this is a very nice paper and I like the choice of experiments. I have just a few comments:

(1) The authors say that: “Existing works that have investigated the benefit of using objects have either assumed that an interface to an idealized object space already exists or that supervision is available to learn a mapping between raw inputs and relevant object properties (for instance, category, position, and orientation).”

The following paper is very relevant and they don’t make either of the assumptions that the authors state in their paper (quoted above). RELATIONAL NEURAL EXPECTATION MAXIMIZATION: UNSUPERVISED DISCOVERY OF OBJECTS AND THEIR INTERACTIONS  -  Steenkiste et al. ICLR 2018. Steenkiste et al. automatically learn to segment objects and predict the physics across multiple time steps. A detailed comparison between the authors' model and that of Steenkiste et al. would make the authors contributions more clear.

(2) Could you please clarify if *ground truth* object segments are fed into the Perception model? If *ground truth* object segments are used, this should be made more clear. (The last line in the caption of Figure 2 is not sufficient to make this clear in the main text).

(3) Very minor, but in Figure 2, the yellow triangle appears to change colour, to green.

(4) How exactly is the model is Figure 2 trained? Is it trained to predict t=1 given t=0? If so how are reconstructions in Figure 3, for t=0 obtained? Is the physics engine bypassed to obtain a reconstruction for t=0? This is not clear. Is the reconstruction (error) for t=0 used to train the model? It is not clear what loss functions are used for training?

(5) Figure 5 and section 4.3 are really nice!

---

> ### Author Response · Authors · 2018-10-03
> **Comparisons and training clarifications**
>
> Thank you for your detailed feedback.
>
> 1. This is a good point. We cite Neural Expectation Maximization (N-EM; Greff et al, 2018) when discussing disentangled object representations, but Relational NEM (R-NEM) is indeed more relevant because it incorporates physical interactions into the model. It is our understanding that the R-NEM code works only on binary images of 2D objects, whereas we consider color images of 3D objects. R-NEM focuses on disentangling objects completely unsupervised, so does not use object segments but is evaluated on simpler inputs. In comparison, we focus on using object representations for downstream tasks, so assume an accurate preprocessing step to give segments but use our object representations in contexts other than prediction (like block stacking).
>
> These works tackle complementary pieces of the same larger problem, and one could imagine a full pipeline using something like R-NEM to discover segments to feed into our method for planning action sequences with learned object representations. We will add this discussion to the next version of the paper.
>
> 2 Yes, we outline the segmented images in orange in Figure 2c because we are using ground truth object segments. We assume we have access to this preprocessing at both train and test time. We will make this more clear in the main text.
>
> 3. The rendered scene has a few forward-facing lights at about two-thirds of the image’s height, so most objects appear a bit brighter before they are dropped. You can also see this happening in Figure 6.
>
> 4. We train the model to reconstruct images at both t=0 and t=1 given the observation at t=0. The loss for the image at t=0 is equation (1) on page 6:
>
>         L_2(\hat{I}_0, I_0) + L_vgg(\hat{I}_0, I_0),
>
> where L_vgg is a perceptual loss in the feature space of the VGG network. The analogous loss for t=1 is equation (2). As you mention, reconstructing the t=0 image essentially amounts to bypassing the physics engine. A more complete description is given in the last paragraph on page 5.
>
> Please let us know if you have any follow-up questions.

---

### Author Response · Authors · 2018-11-28
**Draft update**

We would like to thank the reviewers and commenters for their feedback on our submission. Our revised draft incorporates many of their suggestions. Most importantly:

1. We have run our model and planning procedure on a Sawyer robotic arm using real goal images. Results can be found at the following website: https://sites.google.com/view/object-models
as well as in the new Section 3.4 and Appendix B of the revision. Our results, robot stacking of up to 9 shapes directly from real images, has not been demonstrated in prior work, regardless of the complexity of those shapes.

2. We have given a more precise description of the planning procedure in Algorithm 1 on page 4.

Other changes are discussed in the individual responses below.

---

### Meta-Review · Area_Chair1 · 2018-12-13
**novel approach with good performance on interesting and challenging problem; clarity could be improved**

**Confidence:** 4
**Recommendation:** Accept (Poster)

**Metareview:**

1. Describe the strengths of the paper.  As pointed out by the reviewers and based on your expert opinion.

- The problem is interesting and challenging
- The proposed approach is novel and performs well.

2. Describe the weaknesses of the paper. As pointed out by the reviewers and based on your expert opinion. Be sure to indicate which weaknesses are seen as salient for the decision (i.e., potential critical flaws), as opposed to weaknesses that the authors can likely fix in a revision.

- The clarity could be improved

3. Discuss any major points of contention. As raised by the authors or reviewers in the discussion, and how these might have influenced the decision. If the authors provide a rebuttal to a potential reviewer concern, it’s a good idea to acknowledge this and note whether it influenced the final decision or not. This makes sure that author responses are addressed adequately.

Many concerns were clarified during the discussion period. One major concern had been the experimental evaluation. In particular, some reviewers felt that experiments on real images (rather than in simulation) was needed.
To strengthen this aspect, the authors added new qualitative and quantitative results on a real-world experiment with a robot arm, under 10 different scenarios, showing good performance on this challenging task. Still, one reviewer was left unconvinced that the experimental evaluation was sufficient.

4. If consensus was reached, say so. Otherwise, explain what the source of reviewer disagreement was and why the decision on the paper aligns with one set of reviewers or another.

Consensus was not reached. The final decision is aligned with the positive reviews as the AC believes that the evaluation was adequate.